# Role of reptiles and associated arthropods in the epidemiology of rickettsioses: A one health paradigm

**Jairo Alfonso Mendoza-Roldan**[1], **Ranju Ravindran Santhakumari Manoj**[1], **Maria Stefania Latrofa**[1], **Roberta Iatta**[1], **Giada Annoscia**[1], **Piero Lovreglio**[2], **Angela Stufano**[2], **Filipe Dantas-Torres**[3], **Bernard Davoust**[4,5], **Younes Laidoudi**[4,5], **Oleg Mediannikov**[4,5], **Domenico Otranto**[1,6]*

**1** Department of Veterinary Medicine, University of Bari, Valenzano, Italy, **2** Interdisciplinary Department of Medicine, Section of Occupational Medicine, University of Bari, Bari, Italy, **3** Aggeu Magalhães Institute, Oswaldo Cruz Foundation, Recife, Brazil, **4** IRD, AP-HM, MEPHI, Aix Marseille University, Marseille, France, **5** IHU-Méditerranée Infection, Marseille, France, **6** Faculty of Veterinary Sciences, Bu-Ali Sina University, Hamedan, Iran

\* domenico.otranto@uniba.it

## Abstract

We assessed the presence of *Rickettsia* spp., *Coxiella burnetii* and *Anaplasma phagocytophilum* in reptiles, their ectoparasites and in questing ticks collected in a nature preserve park in southern Italy, as well as in a peri-urban area in another region. We also investigated the exposure to these pathogens in forestry workers, farmers and livestock breeders living or working in the nature preserve park given the report of anecdotal cases of spotted fever rickettsioses. *Rickettsia* spp. were molecularly detected in *Podarcis muralis* and *Podarcis siculus* lizards (i.e., 3.1%), in *Ixodes ricinus* (up to 87.5%) and in *Neotrombicula autumnalis* (up to 8.3%) collected from them as well as in *I. ricinus* collected from the environment (up to 28.4%). *Rickettsia monacensis* was the most prevalent species followed by *Rickettsia helvetica*. An undescribed member of the family Anaplasmataceae was detected in 2.4% and 0.8% of the reptiles and ectoparasites, respectively. Sera from human subjects (n = 50) were serologically screened and antibodies to *Rickettsia* spp. (n = 4; 8%), *C. burnetti* (n = 8; 16%) and *A. phagocytophilum* (n = 11; 22%) were detected. Two ticks collected from two forestry workers were positive for spotted fever group (SFG) rickettsiae. *Ixodes ricinus* is involved in the transmission of SFG rickettsiae (*R. monacensis* and *R. helvetica*) in southern Europe and lizards could play a role in the sylvatic cycle of *R. monacensis*, as amplifying hosts. Meanwhile, *N. autumnalis* could be involved in the enzootic cycle of some SFG rickettsiae among these animals. People living or working in the southern Italian nature preserve park investigated are exposed to SFG rickettsiae, *C. burnetii* and *A. phagocytophilum*.

**Data Availability Statement:** All relevant data are within the manuscript and its Supporting Information files.

**Funding:** The authors received no specific funding for this work.

**Competing interests:** The authors have declared that no competing interests exist.

## Author summary

Zoonotic pathogens such as *Rickettsia* spp., *Coxiella burnetii* and *Anaplasma phagocytophilum* are associated with ticks, lice, fleas and mites and may infect a wide range of vertebrate species. There are still many knowledge gaps concerning the arthropod vectors and amplifying hosts of these pathogens. Reptiles are known to host infected ticks with these zoonotic pathogens and eventually become infected themselves by these bacteria. Hence to address this question from a broad One-Health perspective, we assessed the molecular prevalence of *Rickettsia* spp., *C. burnetii* and *A. phagocytophilum* in reptiles, ticks and mites from both hosts and environment and seroprevalence in humans living or working near the same area. *Rickettsia* DNA was detected in lizards' tails, *Ixodes ricinus* ticks and in *Neotrombicula autumnalis* mites collected from them and in questing ticks. Although DNA of *C. burnetii* and *A. phagocytophilum* was not detected in lizards and their ectoparasites, sera samples of human subjects tested positive for the antibodies against these zoonotic pathogens. These findings indicate the epidemiological role of lizards in spreading SFG rickettsiae as well as *I. ricinus*, and hence representing a potential public health concern in geographical areas where lizards, ticks and humans share the same environment.

## Introduction

Many examples of tick-borne diseases highlight the pivotal role of adoption of a One-Health approach for understanding the ecology of pathogens transmitted by ticks to human and animal populations, living in the same environment [1]. Rickettsioses caused by *Rickettsia* spp. represent a paradigmatic example for that. Similarly, other intracellular bacteria of public health concern, mainly in the USA and Europe, are *Coxiella burnetii* causing coxiellosis/ Q fever and *Anaplasma phagocytophilum* causing human granulocytic anaplasmosis (HGA) [2,3]. These pathogens are associated with arthropod vectors (i.e., ticks, lice, fleas and mites) and may infect a wide range of vertebrate species [4–6]. Pets (i.e., dogs and cats), reptiles, ticks, rodents and birds are known to contribute, at different extent, in the dissemination of these pathogens [3,7]. The epidemiology of rickettsioses, coxiellosis and HGA are intimately related to the corresponding pathogen, arthropod vector, vertebrate host, and the environment where they thrive. While a considerable amount of information on the ecology of these pathogens has been acquired in the last decades [7–10], there are still many knowledge gaps concerning the arthropod vectors and amplifying hosts. In humans, *Rickettsia* spp. infection causes conditions of various degrees of severity according to different factors such as the bacterial species, the individual susceptibility and the immune status [8,11–13]. Mediterranean spotted fever (MSF) by *Rickettsia conorii* causes the most prevalent and severe form of rickettsiosis, in Europe. Typical manifestations in human patients are fever, cutaneous maculo-papular rash, and eschar at the site of tick bite accompanied by regional lymphadenopathies. Sometimes, unspecific flu-like symptoms may be the only manifestations, but severe or lethal outcomes may occur if proper diagnosis is not performed and antibiotic therapy is delayed [14,15]. Rickettsioses caused by *Rickettsia slovaca* and *Rickettsia raoultii* are responsible for so-called SEN-LAT (scalp eschar and neck lymphadenopathy after tick bite) syndrome, also known as TIBOLA (tick-borne lymphadenopathy) or DEBONEL (*Dermacentor*-borne necrosis erythema lymphadenopathy). Other rickettsiae in Europe (e.g., *Rickettsia aeschlimannii*, *Rickettsia helvetica* and *Rickettsia massiliae*) are responsible for less common and, probably, underreported febrile illnesses. Furthermore, *Rickettsia monacensis* was reported in Europe as a cause of MSF-like illness in Italy and Spain [16,17]. The gamma-proteobacterium *Coxiella*

*burnetii* is the causative agent of Q fever, a flu-like disease manifesting from self-limiting non-specific fever to atypical pneumonia, hepatitis, endocarditis and neurological manifestation. HGA causes nonspecific febrile illness, which could lead to a fatal outcome [7,10]. Though, Q fever was primarily considered a regionally restricted zoonotic disease, it has been widely diagnosed in most countries by serological analyses [18].

Most rickettsiae are primarily hosted and transmitted by ticks. *Rickettsia conorii* is usually transmitted by the brown dog tick (*Rhipicephalus sanguineus* sensu lato), and *Rickettsia slovaca* and *R. raoultii* by *Dermacentor* spp. [9], although non-specific carriage was also identified for many rickettsial species, especially when different tick species live in sympatry [19]. *Coxiella burnetii* infections are reported in livestock, domestic and wild mammals, birds and a wide variety of ticks [20]. Although ticks are not considered essential in the natural cycle of this pathogen, it multiplies in the gut cells of tick genera such as *Ixodes*, *Haemaphysalis*, *Rhipicephalus* and *Dermacentor* and is shed in tick faeces [21,22].

Because many tick-borne rickettsiae are primarily associated to ticks infesting wildlife, rickettsioses, coxiellosis and HGA are often diagnosed in people living or working in the proximities of forested areas (e.g., park rangers, foresters and hunters) and occupational farmers, veterinarians, laboratory technicians, slaughterhouses and cheese factories personnel. For instance, seroprevalence of *Rickettsia* and *Coxiella* antibodies may be up to 37% and up to 23.8%, respectively, in this occupational risk group in southern regions of Italy [3,23,24], where tick bites are often reported [25]. In a large survey conducted on human infested by ticks in Italy, *I. ricinus* was the most frequently retrieved species (i.e., 59.5%) carrying the highest number of pathogens, including *Rickettsia* spp. [25]. This tick has a catholic feeding behaviour, being found infesting a wide range of hosts, including reptiles and humans [26]. Meanwhile, *I. ricinus*-associated rickettsiae (e.g., *R. helvetica* and *R. monacensis*) have been detected in synanthropic reptiles, especially lacertid lizards [26,27]. Indeed, reptiles are known to act as hosts of ticks carrying *Rickettsia* spp., *C. burnetii*, *A. phagocytophilum*, *Borrelia lusitaniae* and eventually become infected themselves by these bacteria [7,10,11,26–31,32]. Nonetheless, the significance of reptiles, such as lizards, in the ecology of Rickettsiales and in the epidemiology of human infections is still to be ascertained.

The collection of lizards infested by *I. ricinus* and mites (*Neotrombicula autumnalis*) in woody areas of southern Italy [32], along with anecdotic reports of people complaining of MSF-like clinical signs in the same areas stimulated the present investigation. Our main objective was to detect these rickettsial microorganisms in reptiles, their ectoparasites, and free-living ticks from the same environment. To address this question from a broad One Health perspective, the seroprevalence of antibodies against *Rickettsia*, *C. burnetii* and *A. phagocytophilum* was also assessed in humans living or working near the same woody area.

## Methods

### Ethics statement

The study was conducted in accordance with ethical principles (Declaration of Helsinki), written informed consent was obtained from the participants and the research protocol was approved by the ethics committee of the University Hospital of Bari (n. 6394, prot. n., 0044469–23062020). Protocols for lizard collection and sampling were approved by the Commission for Bioethics and Animal Welfare of the Department of Veterinary Medicine of the University of Bari and authorized by the Ministry for Environment, Land and Sea Protection of Italy (approval number 0016973/2018), the Societas Herpetologica Italica (approval number SHI-aut-ER-12-2018) and the "Istituto Superiore per la Protezione e la Ricerca Ambientale" (approval number 41180).

## Study area and sample collection

The study was conducted from 2018 to 2020 in the Basilicata region in southern Italy in the Gallipoli Cognato Forest (site 1, 40˚32'17"N, 16˚07'20.17"E), which belongs to the Parco Regionale di Gallipoli Cognato e delle Piccole Dolomiti Lucane (27,027 ha in extension), where a great diversity of animals and plants is present. With a vegetation coverage varying according to altitude, this forest of Turkey oak (*Quercus cerris*) is a known habitat for a plethora of parasites, including ticks [33–35].

All reptiles' samples (i.e., blood and tail tissue) and ectoparasites were collected under the frame of previous studies [32,33]. The first site (site 1) from where lizards and ticks were collected was a meadow habitat within an enclosure inhabited by roe deer (*Capreolus capreolus*) (40˚32'17"N, 16˚07'20.17"E). The collection site was bordered by or within the forest, being selected on authors' observations of environmental features (e.g., vegetation cover), as well as the experience in previous studies on tick ecology [33,36]. In brief, ticks (n = 250 *I. ricinus*) were collected from the environment by dragging and flagging as detailed in a previous study [36] and morphologically identified [37,38]. In this location, lizards (n = 128) and snakes (n = 4) were captured and checked for ectoparasites as described elsewhere [32]. In addition, two adult females of *I. ricinus* and one adult male of *Dermacentor marginatus* collected from two park rangers were also screened. The second collection site (site 2) was represented by a peri-urban area in the surroundings of the Department of Veterinary Medicine, University of Bari "Aldo Moro" (41˚1'31.584"N, 16˚54'3.6288"E), in the province of Bari, where lizards (n = 40) were captured and found infested solely by mites [32].

## DNA extraction, polymerase chain reaction (PCR) and phylogenetic analyses

Genomic DNA was extracted from ticks and mites using a lysis with guanidine isothiocyanate protocol (GT) [39], and eluted in AE elution buffer (50 μl for mites and ticks). Whilst, DNA from reptile blood (~20 μl) and from tail tissue (25 mg) of lizards was extracted by using a Qiagen Mini kit and Qiagen DNeasy Blood & Tissue kit (Qiagen, Hilden, Germany), respectively.

DNA samples were tested by PCR using a pair of primers (CS-78F and CS-323R) targeting a fragment (401bp) of the gene citrate synthase (*gltA*), present in all species of *Rickettsia* [40]. The PCR runs were performed in the Master cycler Gradient (Eppendorf California) thermocycler using the following thermal cycling conditions: 95˚ C for 5 minutes, followed by 40 cycles of 95˚ C for 30 seconds, 58˚ C for 30 seconds and 72˚ C for 40 seconds and 72˚ C for 7 minutes. Positive samples were tested by a second PCR using a pair of primers as previously described [41,42] (Rr190.70F and Rr190.701R) targeting a fragment of the outer membrane protein A (*ompA*) gene (632 bp), present only in spotted fever group (SFG) *Rickettsiae*. The cycling conditions for the *ompA* gene were: 95˚ C for 5 minutes, followed by 35 cycles of 95˚ C for 40 seconds, 58˚ C for 30 seconds, 72˚ C for 45 seconds and 72˚ C for 10 minutes. Samples were also screened for *C. burnetii* and for species of the Anaplasmataceae family using primer pairs (CAPI-844-F and CAPI-844-R; EHR16SD and EHR16SR) which amplified a 601 bp fragment of the CAPI gene [43] and 345 bp of 16S rRNA gene [44], respectively. The cycling conditions used for CAPI gene were 95˚ C for 10 minutes followed by 35 cycles of 95˚ C for 45 seconds, 60˚ C for 45 seconds and 72˚ C for 45 seconds and 72˚ C for 7 minutes while that of 16S rRNA gene were 95˚ C for 10 minutes followed by 35 cycles of 95˚ C for 30 seconds, 60˚ C for 30 seconds and 72˚ C for 30 seconds and 72˚ C for 10 minutes. In all PCR runs negative (Milli-Q water) and positive controls of the respective pathogens were included.

Amplified DNA were subjected to electrophoresis in a 2% agarose gel stained with GelRed (VWR International PBI, Milano, Italy) and viewed on a GelLogic 100 gel documentation system (Kodak, New York, USA). Amplicons were purified using 10 μl of PCR product mixed with 0.5 μl of *Escherichia coli* exonuclease I (*Exo*I; MBI, Fermentas, Lithuania), 1 μl of shrimp alkaline phosphatase (SAP) and 0.5 μl of SAP reaction buffer (MBI, Fermentas, Lithuania) to remove unused primers and unincorporated dNTPs. This mix was incubated at 37°C for 20 min, following enzymes inactivation at 85°C for 15 min. PCR purified products were sequenced using the Taq Dye Doxy Terminator Cycle Sequencing Kit (v.2, Applied Biosystems, California, USA) in an automated sequencer (ABI-PRISM 377). Sequences were analysed by Geneious version 11.1.4 software and compared with those available in Genbank database by Basic Local Alignment Search Tool (BLAST) [45].

Rickettsial *ompA* and *gltA* as well as 16S rRNA genes from *Anaplasma* spp. were amplified and the sequences were separately aligned against those closely related species available from GenBank database using the ClustalW application within MEGA6 software [46]. The Akaike Information Criterion (AIC) option in MEGA6 [46] was used to establish the best nucleotide substitution model adapted to each sequence alignment. Tamura 3-parameter model with a discrete Gamma distribution (+G) [47] was used to generate the *ompA*, *gltA* and the 16S rRNA trees. A maximum likelihood (ML) phylogenetic inference was used with 1000 bootstrap replicates to generate the phylogenetic tree in MEGA6 [46]. Homologous sequences of *Rickettsia* were used as outgroup to root the trees, including the *glt*A sequences from *Rickettsia belli* and *Rickettsia canadensis* (JQ664297, AB297809), the *omp*A sequence of *Rickettsia felis* (AY727036) and the 16S sequence of *Rickettsia parkeri* (NR118776).

## Human blood collection and serological testing

In February 2020, serum samples (n = 50) were collected from forestry workers and tour guides operating within site 1, and among farmers and livestock breeders employed in the municipalities included in regional park area. All workers were asked to fill a questionnaire providing socio-demographic characteristics, working history, previous exposure to tick bites as well as medical history with special focus on the three months prior to the study. All the participants were fully informed about the research aims and features and were provided with an informed consent to take part, before filling the questionnaire, in accordance with the Helsinki Declaration (WMA 2013). Blood samples (10 ml) were collected in a Vacutainer tube for each participant and transported at 4°C to the laboratory where serum samples were obtained by centrifugation at 3000 × g for 30 min. Serum samples were stored in 2 ml tubes at -20°C until serological analyses were performed.

All human sera were screened for five selected tick-borne bacteria belonging to the Rickettsiales and Legionellales orders using the quantitative indirect immunofluorescence antibody test (IFAT). IFAT is the reference method in diagnosing these infections [18] which have been proven to be 100% sensitive. First, two-fold dilutions (1:50 and 1:100) of sera (including positive and negative sera) were prepared in phosphate buffered saline (PBS), and a dilution of 1:100 was selected as the cut-off value. Briefly, each slide well was sensitized using in-house generated following antigens: *R. conorii* (strain Malish 7, ATCC VR-613), *R. felis* (strain Marseille, ATCC VR-1525), and *Rickettsia typhi* (strain Wilmington), *R. helvetica* (strain C9P9), phases I and II of *C. burnetii* (Nine Mile RSA493 strain) and *A. phagocytophilum* (Webster strain). The antigens were deposited separately as microdroplets all around the periphery of each well following the same order in all slides. In a second step, each human serum was investigated separately for both IgG and IgM. Twenty microliters of each human serum dilution was applied per well and slides were then incubated for 30 min at 37°C. After incubation, slides

were washed twice with PBS for 5 minutes and once with distilled water. Twenty microliters of mouse anti-human total immunoglobulin (Ig) conjugated with fluorescein isothiacyanate (FITC) (Sigma-Aldrich, St Louis, MO, USA) were added into each well. Slides were immediately incubated at 37˚ C for 30 minutes and then washed following the same procedure described above. Positive sera at the cut-off of 1:100 were further investigated for IgG and IgM using a five 2-fold serial dilution 1:200 to 1: 3400.

## Results

Of the snakes (n = 4; 2 *Elaphe quatuorlineata*, 1 *Hierophis carbonarius*, 1 *Natrix natrix*) and lizards (n = 128; 5 *Lacerta viridis*, 15 *Podarcis muralis*, 106 *Podarcis siculus*, 2 *Tarentola mauritanica*) captured in site 1, 94.7% (125/132) were infested with mites and ticks. In particular, 123 (93.2%) (1 *E. quatuorlineata*, 5 *L. viridis*, 15 *P. muralis*, 102 *P. siculus*) reptiles were infested with *I. ricinus* (Fig 1) and/or *N. autumnalis* (Fig 2). Only lizards (n = 40; 1 *P. muralis*, 29 *P. siculus*, 10 *T. mauritanica*) were captured in site 2, with 37 (92.6%) (1 *P. muralis*, 29 *P. siculus*, 9 *T. mauritanica*) of them being infested with mites and, particularly, 10 *P. siculus* (25%) by *N. autumnalis*. The mean intensity (5.5, 95% CI: 5.4–6.2%) and abundance (5, 95% CI: 4.45–5.6%) were calculated on a total of 120 reptiles from site 1, which were infested with *I. ricinus*.

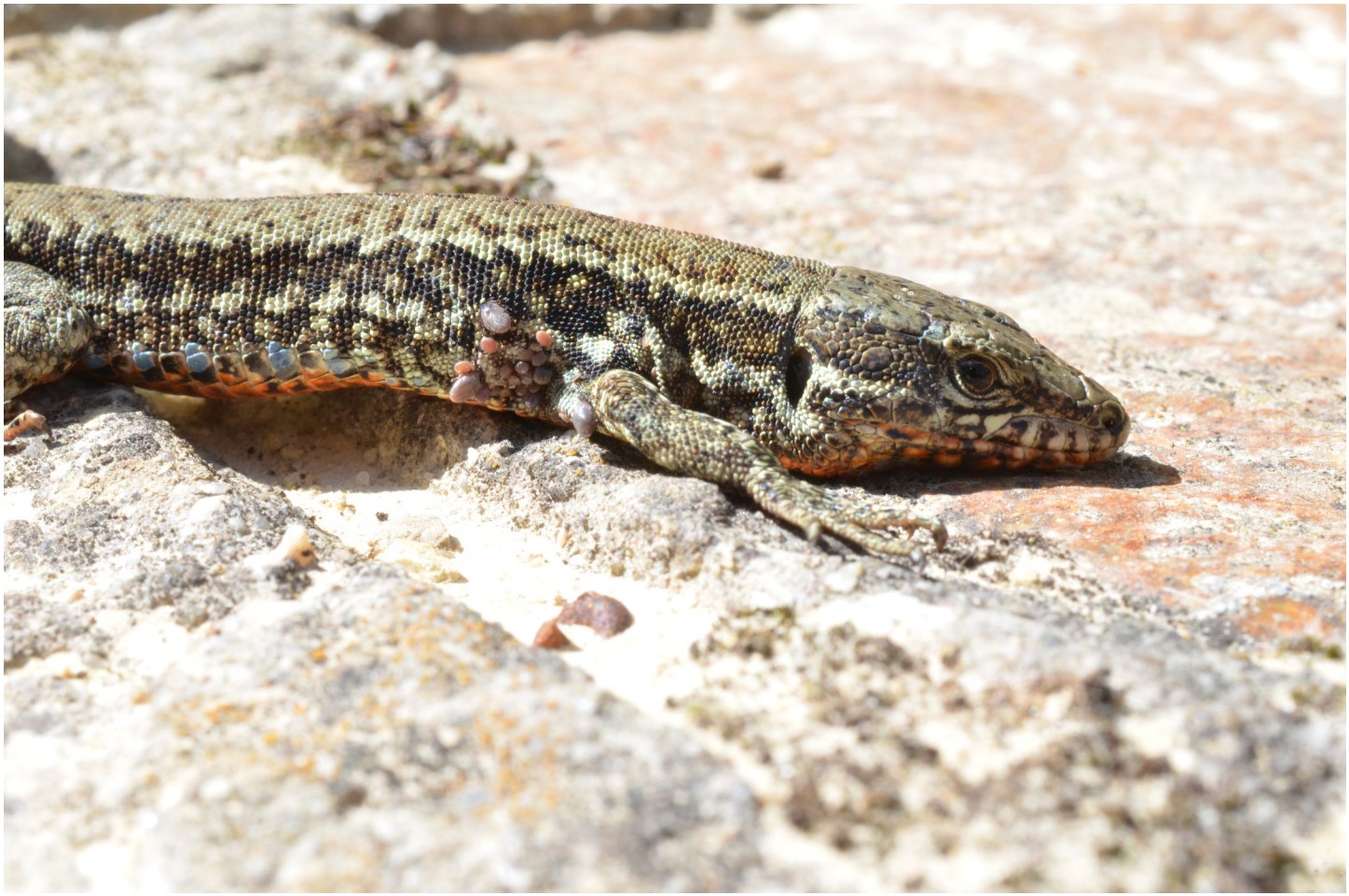

**Fig 1.** *Ixodes ricinus* larvae and nymphs in the axillary region of adult *Podarcis muralis*.

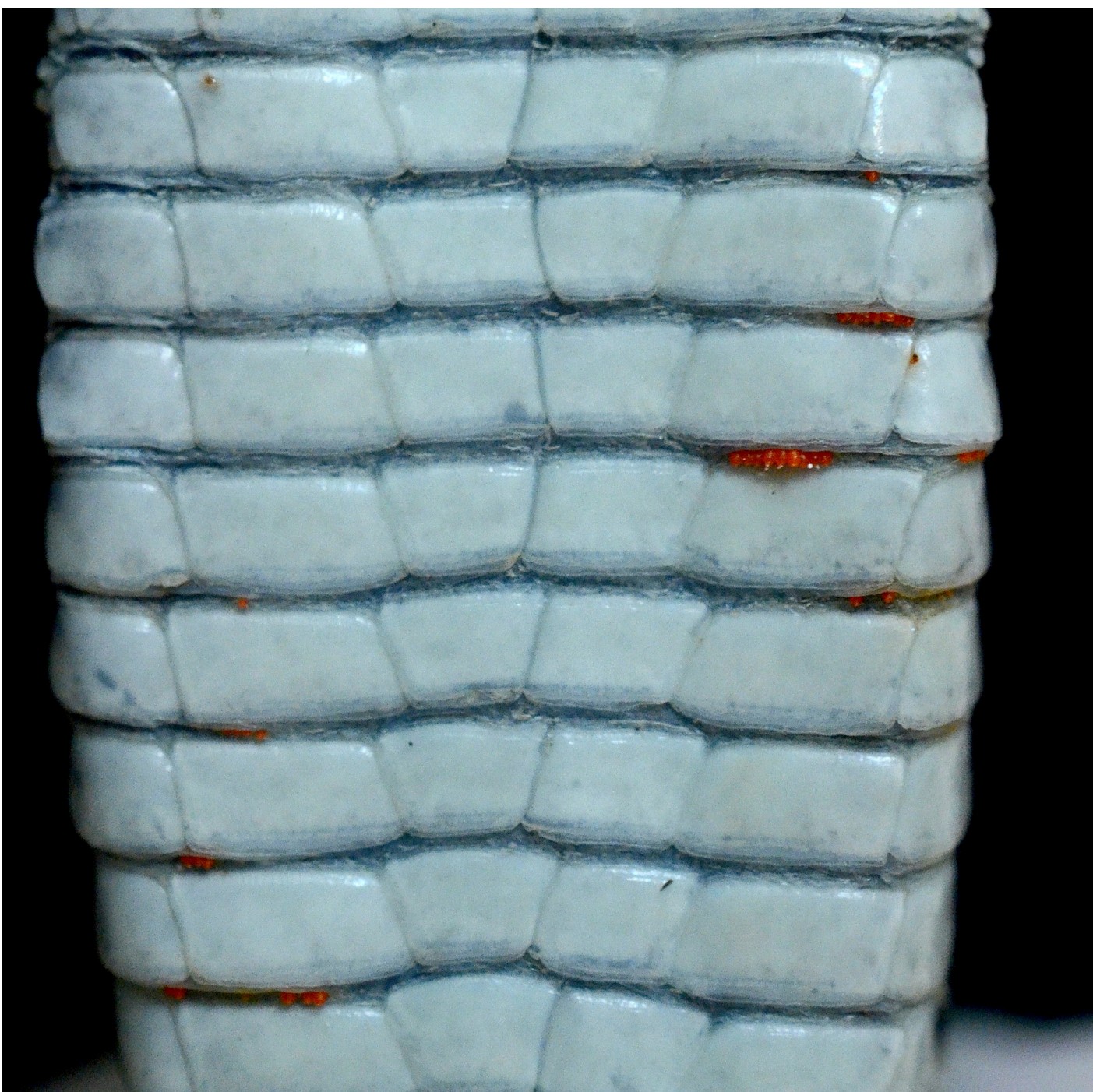

**Fig 2.** *Neotrombicula autumanlis* **larvae in the ventral region of adult** *Podarcis siculus.*

Of the 172 reptile samples, Rickettsiales DNA was assessed molecularly from 158 (91.9%) infested with *I. ricinus* and/or *N. autumnalis*.

The *Rickettsia gltA* gene was amplified from tail tissues of four (i.e., 3.1%; 95% CI: 0.6–5.8%) lizards (i.e., one *P. muralis* and two *P. siculus* from site 1; one *P. siculus* from site 2). *Rickettsia gltA* gene sequences from specimens from site 1 displayed 100% of nucleotide identity

**Table 1. Number and percentage of ectoparasites (*Ixodes ricinus* and *Neotrombicula autumnalis*) positive for *Rickettsia* spp. collected on reptile hosts.** Percentage of sequence identity for *gltA* and *ompA* genes with sequences available in GenBank.

| Host (infested/total) | Hosts infested by ectoparasite species % (infested/total) | gltA (n positive/total) sequence identity | ompA (n positive/total) sequence identity |
|---|---|---|---|
| *Podarcis siculus* (Italian wall lizard) (112/135) | *Ixodes ricinus* 88.3 (99/112) | (64/99) *Rickettsia monacensis* 100% (KU586332) (24/99) *Rickettsia helvetica* 100% (KU310588) | (64/99) *Rickettsia monacensis* 100% (MK922661) |
| | *Neotrombicula autumnalis* 77.7 (87/112) | (5/87) *Rickettsia monacensis* 100% (KU586332) | (2/5) *Rickettsia monacensis* 100% (MK922661) |
| *Podarcis muralis* (common wall lizard) (15/16) | *Ixodes ricinus* 100 (15/15) | (8/15) *Rickettsia monacensis*100% (KU586332) (3/15) *Rickettsia helvetica* 100% (KU310588) | (8/15) *Rickettsia monacensis* 100% (MF383610) |
| | *Neotrombicula autumnalis* 33.3 (5/15) | (2/15) *Rickettsia monacensis* 100% (KU586332) | (2/15) *Rickettsia monacensis* 100% (MK922661) |
| *Lacerta bilineata* (western green lizard) (5/5) | *Ixodes ricinus* 100 (5/5) | (5/5) *Rickettsia monacensis* 100% (KU586332) | (5/5) *Rickettsia monacensis* 100% (MK922661) |
| | *Neotrombicula autumnalis* 80 (4/5) | (1/4) *Rickettsia monacensis* 100% (KU586332) | |
| *Elaphe quatuorlineata* (four-lined snake) (1/2) | *Ixodes ricinus* 100 (1/1) | (1/1) *Rickettsia helvetica* 100% (KU310588) | |

with that of *R. monacensis* (KU586332), whereas sequence from site 2 was 100% identical to that of *R. helvetica* (KU310588). None of the four snakes tested positive for *Rickettsia* spp.

In addition, 87.5% (105/120; 95% CI: 80.2–92.8%) of *I. ricinus* collected from reptiles were positive for the *Rickettsia gltA* gene, of which 65.8% (79/120; 95% CI: 56.6–74.3%) also for the *Rickettsia ompA* gene. Out of 96 *N. autumnalis* larvae, eight (8.3%; 95% CI: 38.3–56.8%) yielded positive results for the *Rickettsia gltA* gene and four (4.2%; 95% CI: 1.1–10.3%) for the *Rickettsia ompA* gene (Table 1). *Rickettsia monacensis* was the only rickettsial species identified using the *ompA* gene in both ticks and mites of lacertids.

Of the 250 questing *I. ricinus* collected from the environment, 28.4% (71/250; 95% CI: 25.7–30.1%) yielded positive results for the *gltA* gene and 8.8% (22/250; 95% CI: 3.4–7.1%) for the *ompA* gene. Two different *Rickettsia* species, namely *R. helvetica* (9/71; only for the *gltA* gene) and *R. monacensis* (62/71) were identified with 100% nucleotide identity with GenBank reference sequence KU310588 and MK922661, respectively. Of the three ticks collected from park rangers, two scored positive for *Rickettsia ompA* gene, with *I. ricinus* for *R. monacensis* (100% nucleotide identity with MK922661) and *D. marginatus* for *R. slovaca* (100% nucleotide identity with MH532257).

None of the reptiles and ectoparasites scored positive for *A. phagocytophilum* and for *C. burnetii*. Yet, an undescribed member of the family Anaplasmataceae, originally designated as *Candidatus* Cryptoplasma sp. was detected in four lizards (n = 3 blood, n = 1 tail) and in ectoparasites 1 mite and 3 ticks (nucleotide identity ranging from 99.8% to 100% with MG924904; see Discussion). In addition, one tick (from lizard) scored positive for *Ehrlichia* sp. (99.3% nucleotide identity with *E. canis*, MN922610) and a lizard to the same undescribed microorganism (99.0% nucleotide identity with MG924904 and GU734325).

*Rickettsia gltA* sequences obtained from ticks and reptiles clustered within the same clades of *R. monacensis* and *R. helvetica*, respectively, as distinct paraphyletic clades with the exclusion of the other *Rickettsia* spp. (Fig 3A). Similarly, *Rickettsia ompA* sequences clustered with *R. monacensis* and *R. slovaca*, supported by high bootstrap values (i.e., 93%; Fig 3B). The 16S

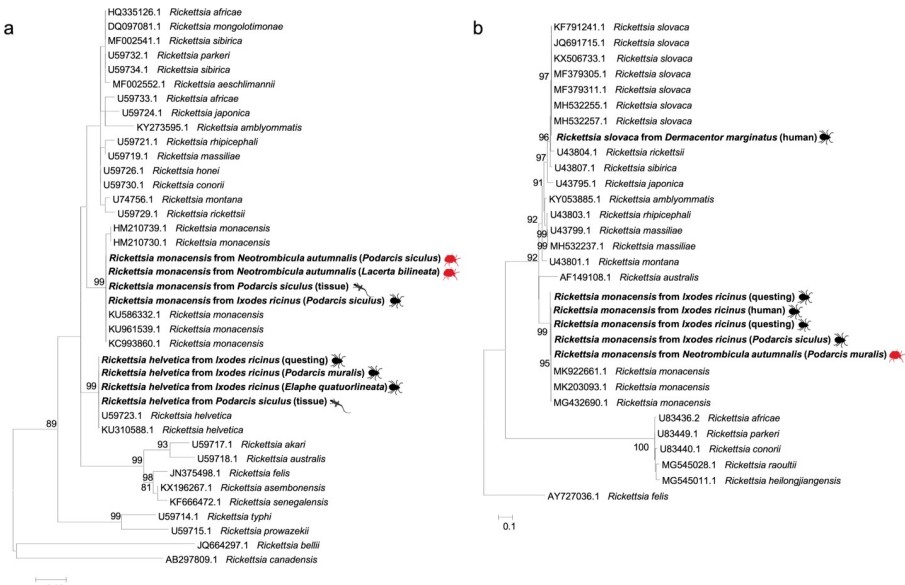

**Fig 3.** Maximum-likelihood phylogenetic trees of *gltA* (A) and *ompA* (B) genes of *Rickettsia* spp. Bootstrap values (>70%) are shown near the nodes. *Rickettsia belli*, *Rickettsia canadensis* (A) and *Rickettsia felis* (B) were used as outgroups. Scale bar indicates nucleotide substitution per site. *Rickettsia* spp. sequenced in this study are in bold. Further details on the origin of samples are in brackets.

rRNA gene sequences clustered with those from a group of undescribed organisms belonging to the family Anaplasmataceae, with the exception of a sequence that clustered within the clade of *Ehrlichia* spp. (Fig 4). Outgroups were consistently resolved as paraphyletic species for both trees. Representative sequences herein generated have been deposited in GenBank under accession numbers from MT829281 to MT829303.

Of the 50 human subjects (Table 2), 18 (36%) were exposed to at least one pathogen of whom four individuals had an IgG titre for more than one pathogen (i.e., 1 for *C. burnetii*, *R. typhi* and *A. phagocytophilum*; 1 for *R. felis* and *A. phagocytophilum*; 2 for *C. burnetii* and *A. phagocytophilum*). The remaining 14 patients showed IgG against *A. phagocytophilum* (n = 7), *C. burnetii* (n = 5), *R. felis* (n = 1) and *R. conorii* (n = 1). Of the 50 human serum samples, four had IgG titres against *Rickettsia* spp.: two to *R. felis* (titres 1:100 and 1:200), one to *R. conorii* (1:100) and another to *R. typhi* (1:400). The last serum also was found reactive when tested with *R. helvetica* antigen with, however, lower titre (1:100). Eleven sera contained antibodies against *A. phagocytophilum* (titres 1:100 to 1:800). No patients had IgM that exclude acute or recent infection. Eight sera reacted with *C. burnetii* antigens (titres 1:200 to 1:6400). No IgM was detected which excluded acute Q fever. However, two patients showed high antibody titres against Phase I of *C. burnetii* (one of 1:1600 and another of 1:6400), with >98% of predictive value suggestive of chronic Q fever.

## Discussion

Results obtained in this study suggest that at least two tick-associated *Rickettsia* spp. (*R. helvetica* and *R. monacensis*) circulate in a lizard population from southern Italy (site 1) as well as in ectoparasites collected on them (i.e., *I. ricinus* and *N. autumnalis*) and in questing ticks collected from the same area. The overall picture of pathogen circulation in that area is also confirmed by the seroprevalence recorded in humans with a previous history of tick bites, as well as by the molecular detection of *Rickettsia* spp. in ticks collected from humans coming from

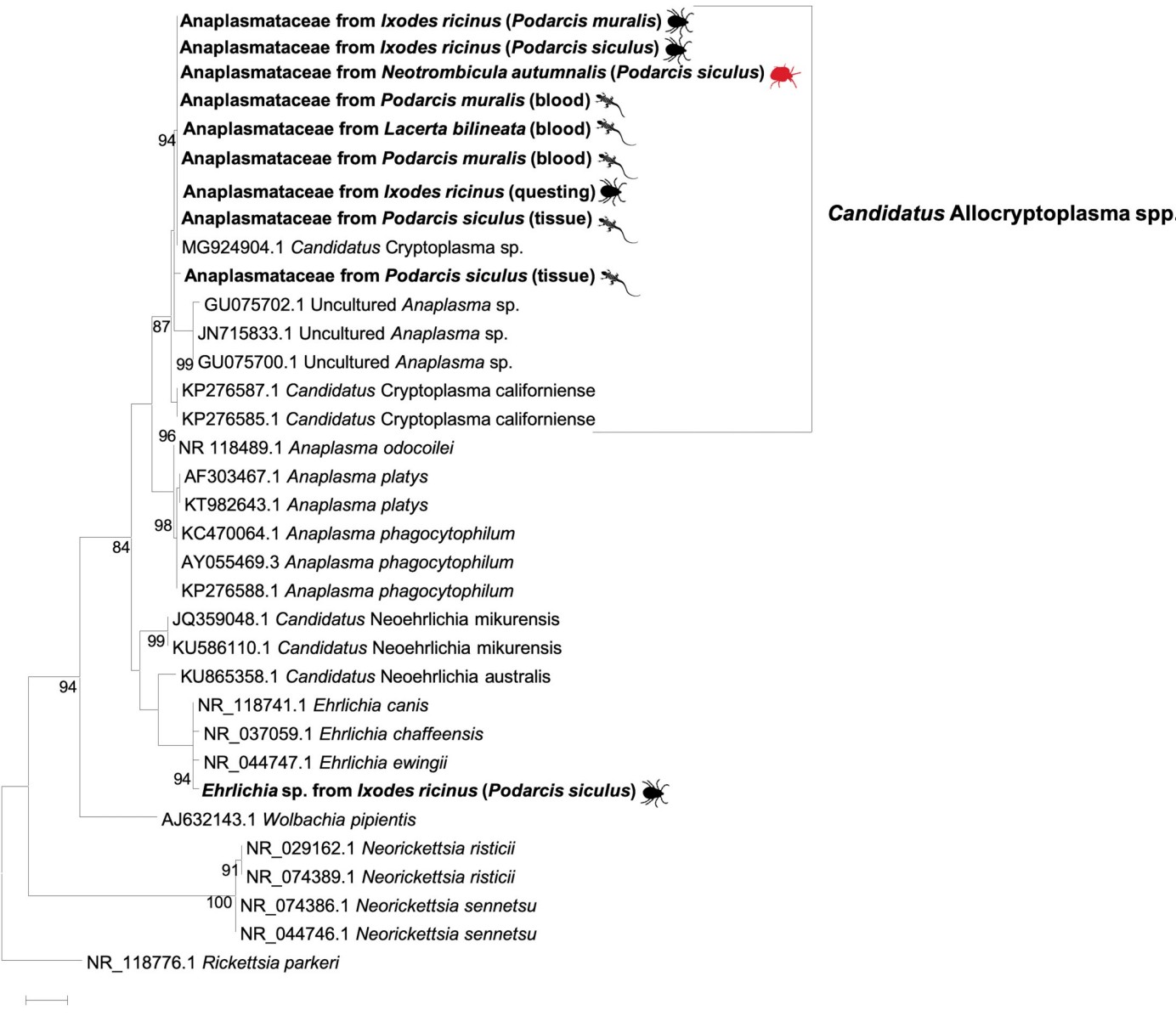

**Fig 4. Maximum-likelihood phylogenetic tree of 16S rRNA sequences of Anaplasmataceae.** Bootstrap values (>70%) are shown near the nodes. *Rickettsia parkeri* was used as outgroup. Scale bar indicates nucleotide substitution per site. Anaplasmataceae sequenced in this study are in bold. Further details on the origin of samples are in brackets.

the same location. Though other animal species (e.g., mice) may also act as hosts of *Rickettsia* spp. in this area, the prevalence of *R. helvetica* and *R. monacensis* in lizards (i.e., 3.1%) and in ticks collected on them (i.e., 87.5%) suggest that these reptiles could be acting as amplifying hosts for these bacteria. Indeed, this is also supported by the detection of rickettsiae in their blood. Present findings agree with other studies in which *R. helvetica* was detected in *P. muralis* tissue (6.2%; 1/16) from the northern Apennine area of Italy [26], and *R. monacensis* in 6.6% (10/151) *Teira dugesii* (Madeiran wall lizard) lizards' tails in Portugal [27]. Since most ectoparasites attach and feed around the axillary region of lizards [31], the detection of *R. monacensis* and *R. helvetica* from the tail tissue may indicate a disseminated infection [26,27]. Low

**Table 2. Anamnestic data on studied human population (age, sex, profession, tick exposure, location).**

| General Characteristics | Exposed workers | | |
|---|---|---|---|
| | Male (n = 41) N (%) Median Range | Female (n = 9) N (%) Median Range | Total (n = 50) N (%) Median Range |
| **Age** | 40.7 21.4–69.3 | 33.5 22.7–59.7 | 39.2 21.4–69.3 |
| **Job** | | | |
| Tour guide | 4 (9.8) | 1 (11.1) | 5 (10) |
| Animal breeder | 6 (14.6) | 0 (0) | 6 (12) |
| Farmer | 1 (2.4) | 1 (11.1) | 2 (4) |
| Forestry workers | 10 (24.4) | 5 (55.6) | 15 (30) |
| Farmer/Animal breeder | 18 (43.9) | 2 (22.2) | 20 (40) |
| Veterinary surgeon | 2 (4.9) | 0 (0) | 2 (4) |
| **Frequency tick bites** | | | |
| Never | 2 (4.9) | 1 (11.1) | 3 (6) |
| Rarely | 9 (22) | 4 (44.4) | 14 (28) |
| Occasionally | 11 (26.8) | 3 (33.3) | 14 (28) |
| Frequently | 19 (46.3) | 1 (11.1) | 19 (38) |
| **Latest tick bite** | | | |
| Never | 2 (4.9) | 1 (11.1) | 3 (7.3) |
| < 7 days | 4 (9.8) | 0 (0) | 4 (9.8) |
| < 1 month | 3 (7.3) | 2 (22.2) | 5 (10) |
| < 1 year | 20 (48.8) | 6 (66.7) | 26 (43.6) |
| < 5 years | 12 (29.3) | 0 (0) | 12 (29.3) |
| **Municipalities** | | | |
| Accettura | 11 (26.8) | 1 (11.1) | 12 (24) |
| Campomaggiore | 5 (12.2) | 1 (11.1) | 6 (12) |
| Pietrapertosa | 4 (9.8) | 1 (11.1) | 5 (10) |
| Oliveto Lucano | 5 (12.2) | 0 (0) | 5 (10) |
| San Mauro Forte | 5 (12.2) | 2 (22.2) | 7 (14) |
| Tricarico | 7 (17) | 2 (22.2) | 9 (18) |
| Laurenzana | 2 (4.9) | 1 (11.1) | 3 (6) |
| Castelmezzano | 2 (4.9) | 1 (11.1) | 3 (6) |

prevalence of *Rickettsia* spp. in reptiles could be explained by the usually short (i.e., for a few days or weeks), transient rickettsiemia in most vertebrates [48]. Like other small vertebrates, lizards are suitable hosts for immature stages of *I. ricinus* ticks across Europe [26,49,50] as recorded in *L. agilis*, *P. muralis*, and *Lacerta viridis* that contribute to the maintenance of *I. ricinus* populations in urban settings [51,52]. Therefore, other species of lizards could potentially act as amplifying hosts for *Rickettsia* spp. Accordingly, based on the retrieval of *R. helvetica* from ticks collected on a snake (*E. quatuorlineata*), the role of ophidian hosts deserves further investigations in order to confirm previous reports of *Rickettsia* spp. DNA in ticks collected from snakes [11]. Conversely, many wild (e.g., wild boars, sika deer, hedgehogs, wild rabbits, opossums, bats, rodents, bandicoots and shrews) and domestic (e.g., cattle, dogs, goat, sheep) mammals could also act as hosts and contribute to the dispersion of *Rickettsia*-infected ticks in different geographical areas [9,53–57]. This is corroborated by the reports of *R. helvetica* DNA in whole blood from mice, roe deer and wild boar [58], which also are animal species present in the studied area. In above-mentioned cases, the amplifying host status of these vertebrates for *Rickettsia* spp. needs confirmatory evidence, such as their natural susceptibility (e.g., antibody detection, *Rickettsia* isolation in culture) and transmissibility of these infectious agents to suitable vectors (i.e., xenodiagnosis) [59].

Under these circumstances, the occurrence of co-feeding of both infected and uninfected ticks could facilitate the circulation of *Rickettsia* spp., such as demonstrated for *R. helvetica* transmission in birds to feeding and co-feeding *I. ricinus* [60]. The detection of *R. helvetica* and *R. monacensis* in *I. ricinus* (free living, from reptiles and from humans), along with the high abundance of this tick in the study area [36], provides a strong circumstantial evidence of

its involvement in the transmission of these rickettsiae in this part of Europe. The occurrence of *Rickettsia* spp. in the study area was initially confirmed in ticks collected from humans, with prevalence up to 17% [25]. However, data on tick-borne rickettsiosis in southern Italy is scarce. Indeed, although spotted fever rickettsiosis is a mandatory notifiable disease in Italy since 1990, its incidence is still unknown due to the hindrances in a specific clinical and serological diagnosis [15].

Considering that larval stages of *N. autumnalis* parasitize reptiles, birds and humans [61], the retrieval of *Rickettsia* spp. in these mites suggests the possibility of the transmission pathways between reptiles and humans. The role of chigger mites as a potential vector of *Borrelia burgdorferi* sensu lato has been previously hypothesized [61]. Our data is of ecological interest considering that only *N. autumnalis* mites were collected on lizards in site 2, where *Rickettsia* spp. were detected in mites and in one lizard. This raises the question whether this could represent a potential threat for human health, even in absence of *I. ricinus*. In any case, considering the biology of *N. autumnalis*, further study about the transstadial and transovarial transmission of *Rickettsia* spp. in this mite are needed to draw further conclusions on this matter.

Overall, *R. monacensis* was the most prevalent species among all screened samples, which is in line with previous studies conducted in Tuscany [62] and Emilia Romagna, Italy [63]. This species, along with other rickettsiae (e.g. *R. massiliae*, *R. aeschlimannii* and *Rickettsia sibirica mongolitimonae*), are considered as emerging human pathogens [64,65]. Accordingly, serological analysis of patients with a history of tick bites revealed antibody titres against *Rickettsia* spp. confirming the risk of human exposure in the study area.

The phylogenetic analyses of the sequences of *R. helvetica* and *R. monacensis* obtained from reptiles, ticks and mites parasitizing them as well as ticks collected on two individuals and questing ticks, revealed the clustering of these *Rickettsia* spp. with that from humans available in GenBank. The high similarity of *Rickettsia* spp. among all the samples indicates the circulation of two sequence types in the studied area, which ultimately gives a hint about the possible role of these crawling creatures for human rickettsial infection. Regardless the possible role of *N. autumnalis* in the ecology of *Rickettsia* spp., public awareness about the risk of mite and tick bites is advocated.

Negative results of PCR from reptiles and their ectoparasites for *C. burnetii* and *A. phagocytophilum* could be due to the more predominant role of the above-mentioned mammalian vertebrate hosts in the ecology and maintenance of these bacteria [66] than that played by reptiles [67]. For instance, rodents have been recorded as feasible reservoirs for *C. burnetii*/Q fever in central Italy [68], as well as ruminants (e.g., cattle, sheep, buffaloes) throughout the country [69]. Whereas, birds and mammals have been indicated as main reservoirs for *A. phagocytophilum* [70]. The high seroprevalence against *A. phagocytophilum* (22%, 11/55) indicates a frequent contact of high-risk group local population with this bacterium, as well as, high seroprevalence for Q fever, with at least two chronic cases.

Finally, the detection of an undescribed member of the Anaplasmataceae in lizards agrees with previous reports from Europe (Slovakia) [71] and USA [72]. This microorganism is related to genus *Anaplasma*, but represents a lineage distinct from all known *Anaplasma* spp. The putative genus "Cryptoplasma" was informally created to accommodate a microorganism originally designated as "*Candidatus* Cryptoplasma californiense" [72], but recently this genus was corrected to "*Allocryptoplasma*" [73]. Still, this genus has not been validly published as yet (as of 27 July 2020). Our results shed new light on the reptile-tick-Anaplasmataceae interactions in Italy and suggest that lizards of the genus *Podarcis* could act as primary hosts for the maintenance and enzootic circulation of undescribed organisms of unknown pathogenicity and zoonotic potential. In the same way, along with a previous study [11], our data suggest that the diversity of ehrlichial microorganisms infecting reptiles is presently underestimated.

In conclusion, data presented suggest that lizards (i.e., *P. muralis* and *P. siculus*) may play a role in the spreading of SFG rickettsiae and that *I. ricinus* is involved in the transmission of these pathogens in southern Europe. Remarkably, lizards could act as amplifying hosts and *N. autumnalis* could be involved in the enzootic cycle of some SFG rickettsiae among these animals. Being *I. ricinus* able to parasitize humans, it may represent a potential public health concern in geographical areas where lizards, ticks and humans live in sympatry. Results herein presented advocate for a One-Health approach to assess the interactions between hosts (including humans), ectoparasites, pathogens and the environment they inhabit, that will aid efficient public health policies in specific epidemiological contexts.

## Acknowledgments

The authors are grateful to Egidio Mallia veterinary responsible of the Parco Regionale di Gallipoli Cognato e Piccole Dolomiti Lucane in Basilicata and the park rangers/workers and farmers for their support in the sampling procedures and their participation in this study. Authors would also like to thank Marcelo Molento for his collaborative efforts during sampling collection.

## Author Contributions

**Conceptualization:** Jairo Alfonso Mendoza-Roldan, Filipe Dantas-Torres, Domenico Otranto.

**Data curation:** Ranju Ravindran Santhakumari Manoj, Roberta Iatta, Filipe Dantas-Torres, Bernard Davoust, Domenico Otranto.

**Formal analysis:** Jairo Alfonso Mendoza-Roldan, Ranju Ravindran Santhakumari Manoj, Maria Stefania Latrofa, Piero Lovreglio, Bernard Davoust, Oleg Mediannikov.

**Investigation:** Jairo Alfonso Mendoza-Roldan, Angela Stufano, Filipe Dantas-Torres, Younes Laidoudi, Oleg Mediannikov, Domenico Otranto.

**Methodology:** Jairo Alfonso Mendoza-Roldan, Ranju Ravindran Santhakumari Manoj, Roberta Iatta, Piero Lovreglio, Angela Stufano, Bernard Davoust, Younes Laidoudi, Oleg Mediannikov, Domenico Otranto.

**Project administration:** Jairo Alfonso Mendoza-Roldan.

**Software:** Maria Stefania Latrofa, Giada Annoscia.

**Supervision:** Domenico Otranto.

**Validation:** Jairo Alfonso Mendoza-Roldan, Maria Stefania Latrofa, Roberta Iatta, Oleg Mediannikov.

**Visualization:** Jairo Alfonso Mendoza-Roldan.

**Writing – original draft:** Jairo Alfonso Mendoza-Roldan, Ranju Ravindran Santhakumari Manoj, Filipe Dantas-Torres, Domenico Otranto.

**Writing – review & editing:** Jairo Alfonso Mendoza-Roldan, Ranju Ravindran Santhakumari Manoj, Maria Stefania Latrofa, Roberta Iatta, Giada Annoscia, Piero Lovreglio, Angela Stufano, Filipe Dantas-Torres, Bernard Davoust, Younes Laidoudi, Oleg Mediannikov, Domenico Otranto.

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
