## [Decision Letter · Decision Letter 0]

19 Oct 2020

Dear Professor Otranto,

Thank you very much for submitting your manuscript "Role of reptiles and associated arthropods in the epidemiology of rickettsioses: a one health paradigm" for consideration at PLOS Neglected Tropical Diseases. As with all papers reviewed by the journal, your manuscript was reviewed by members of the editorial board and by several independent reviewers. In light of the reviews (below this email), we would like to invite the resubmission of a significantly-revised version that takes into account the reviewers' comments. 

We cannot make any decision about publication until we have seen the revised manuscript and your response to the reviewers' comments. Your revised manuscript is also likely to be sent to reviewers for further evaluation.

Sincerely,

Angela Monica Ionica, Ph.D.

Associate Editor

Job Lopez

Deputy Editor

Reviewer's Responses to Questions

**Key Review Criteria Required for Acceptance?**

**Methods**

-Are the objectives of the study clearly articulated with a clear testable hypothesis stated?

-Is the study design appropriate to address the stated objectives?

-Is the population clearly described and appropriate for the hypothesis being tested?

-Is the sample size sufficient to ensure adequate power to address the hypothesis being tested?

-Were correct statistical analysis used to support conclusions?

-Are there concerns about ethical or regulatory requirements being met?

Reviewer #1: Needs to better describe the animals per species/number/ sex, adult and juveniles also give number of ticks and mites collected per animal

for humans: indicate from where were the 50 persons, sex age, profession.

need to indicate to which group the two persons with ticks on them belonged to.

Reviewer #2: (No Response)

Reviewer #3: Methods are clear and state of the art.

**Results**

-Does the analysis presented match the analysis plan?

-Are the results clearly and completely presented?

-Are the figures (Tables, Images) of sufficient quality for clarity?

Reviewer #1: we need a table that give more information on animals per species, age sex and location as well as number of ectoparasites collected on them. (can be expanded from Table 1)

Same for humans (age, sex, location, profession) and who was infested with ticks (sex, age, location)

Reviewer #2: (No Response)

Reviewer #3: Results obtained are clear but low in number and are statistically on weak background

**Conclusions**

-Are the conclusions supported by the data presented?

-Are the limitations of analysis clearly described?

-Do the authors discuss how these data can be helpful to advance our understanding of the topic under study?

-Is public health relevance addressed?

Reviewer #1: overall. discussion is acceptable. May be give some more information on previous report of Coxiella and Anaplasma in reptiles (Publications by Nieto et al., 2009 and Foley et al., 2016).

Reviewer #2: (No Response)

Reviewer #3: It is difficult to draw any conclusion by teh low sample numbers investigated in this study.

**Editorial and Data Presentation Modifications?**

Reviewer #1: Line29: in a nature preserve park

Line 30: in an other Italian region.

Line 33: delete tails as it is confusing. You described it in your M $ M. Same for line 55

Line 38: antibodies anti--just antibodies to 

Line 39: be more precise: Two ticks collected from 2 forestry workers

Line in the southern Italian Nature preserve park (delete herein)

Line 56: in questing ticks. delete from the environment

Line 57: althoug...lizards (plural) Line 58: delete were

Line 68: the USA

Line 71: are known to

Line 73: confusing, as Coxiella is not a Rickettsiae

Line 78 delete the before Mediterranean

LIne 81: site rather than place

Line 86: also known as TIBOLA.

Line 92: cite hepatitis before endocarditis, as hepatitis is usually during acute form and endocarditis in the chronic form.

Line 93: what do you mean with Q fever being regionally restricted?? it is a very common disease everywhere, also more frequent in sheep/goat farming areas.

Line 107: delete anti-

Line 108: in southern region of?? Italy?

Line 118: the collection rather than retrieval

Line 128: the study was..

Line 144: respectively not necessary

Line 151: need to better indicate the numbers. Was only one tick of 1 species collected from tow park rangers? 

it is my understanding that you had 2 ticks total collected on 2 persons? is it correct?

Line 202: need to give, sex, age profession for your 50 individuals.

Lines 203, 204, 206: delete the before site and workers.

Line 219: Phases I and II

Results: numbers do not add with Table 1: I counted 168 animals for sites 1 and 2

You have 4 snakes Line 232 and 2 in Table (Line 251)

It will be useful to know how many ticks were collected per snake/lizard or at list give range and mean).

Replace by by "with" lines 233 and 235

Line 240: None of the four snakes...

Line 245: the only Rickettsial species identified using the ompA

Give the common names for the lizards: P. siculus: Italian wall lizard

P muralis: common wall lizard

Lacerta: Western green lizard

Elaphe: four line snake)

Lines 253-255: How many were positive for both genes?

Line 255: thwo different Rickettsia species

Line 257: The two ticks..

Lines 261 and 264: which species?

Line 276: replace patients by subjects of whom four individuals

Line 287: suggestive of chronic Q fever. Reference 48 should be in the discussion.

Line 291: in a lizard population from Southern Italy (site 1)

Line 295: on them?? you mean humans?

Line 299: Present findings (delete herein presented

Line 301: Madeiran wall lizard (Teira dugesii)

Line 315: which also are

Line 316:delete "in turn"

Line 321 transmission in birds...

Line 343: from reptiles, ticks, and mites ...as well as ticks collected on two individuals...

Lines 347-348: This statement is out of context, as you indicate number of pet reptiles, but your study is on wild free-ranging reptiles in Italy. You should delete this sentence.

For Anaplasma and Coxiella, you need to indicate that it is very uncommon in reptiles....data available from Nieto et al., 2009 and Foley et al., 2009

Reviewer #2: (No Response)

Reviewer #3: P2L42: the role of N. autumnalis is rather vague in this respect, therefor I would omit this from the abstract.

P2L48 : Anaplasma phagocytophilum

P3L52: “One-Health”

P3L54: Anaplasma phagocytophilum

P3L64: rephrase “how pivotal is…”

P3L69: Anaplasma phagocytophilum

P4L80: “… in Europe. Typical manifestations in human patients are …”

P4L90: “The Gamma- …”

P4l101: “…and is shed..”

P5L108: please clarify: “.. regions of “

P5L114: I miss Borrelia lusitaniae

P6L138: what time period

P7L159 bracket is missing.

P7L166: space befor 72°C

P8L188ff: genbank is registered trademark ®

P9L213. Rickettsiales and Legionellales

P10L243: Maybe some statistics showing the relation eg. Kappa could help to understand the relations

P10l248 table 1: for me the table is confusing and I cannot understand the sense of it. It is not possible to distinguish the pathogens in respect to the different hosts (which line belongs to which host or vector…?)

P14L330: N. autumnalis as potential vector is really vague and not proofen. Furthermore the biology of this larval parasitism without any data on transstadial and transovarial transmission studies contradicts any possible transmission and therefor has to be handled with caution. 

P16L376 “One-Health”

**Summary and General Comments**

Reviewer #1: an interesting study which brings important new information on animals that are not been specifically addressed (reptiles and their ticks).

need to better describe each group of animals or humans

Reviewer #2: (No Response)

Reviewer #3: The integral view on this topic clearly merits publication, although the major drawback is the low sample number in all respective material. The data aswell as the outcome does not justify that long list of authors and full paper.

PLOS authors have the option to publish the peer review history of their article (what does this mean?). If published, this will include your full peer review and any attached files.

Reviewer #1: No

Reviewer #2: No

Reviewer #3: No
---

## [Decision Letter · Decision Letter 1]

29 Dec 2020

Dear Proffesor Otranto,

We are pleased to inform you that your manuscript 'Role of reptiles and associated arthropods in the epidemiology of rickettsioses: a one health paradigm' has been provisionally accepted for publication in PLOS Neglected Tropical Diseases.

Best regards,

Angela Monica Ionica, Ph.D.

Associate Editor

Job Lopez

Deputy Editor

Reviewer's Responses to Questions

**Key Review Criteria Required for Acceptance?**

**Methods**

-Are the objectives of the study clearly articulated with a clear testable hypothesis stated?

-Is the study design appropriate to address the stated objectives?

-Is the population clearly described and appropriate for the hypothesis being tested?

-Is the sample size sufficient to ensure adequate power to address the hypothesis being tested?

-Were correct statistical analysis used to support conclusions?

-Are there concerns about ethical or regulatory requirements being met?

Reviewer #1: acceptable

Reviewer #3: (No Response)

**Results**

-Does the analysis presented match the analysis plan?

-Are the results clearly and completely presented?

-Are the figures (Tables, Images) of sufficient quality for clarity?

Reviewer #1: acceptable

Reviewer #3: (No Response)

**Conclusions**

-Are the conclusions supported by the data presented?

-Are the limitations of analysis clearly described?

-Do the authors discuss how these data can be helpful to advance our understanding of the topic under study?

-Is public health relevance addressed?

Reviewer #1: acceptable

Reviewer #3: (No Response)

**Editorial and Data Presentation Modifications?**

Reviewer #1: none

Reviewer #3: (No Response)

**Summary and General Comments**

Reviewer #1: acceptable

Reviewer #3: All suggestions were done

PLOS authors have the option to publish the peer review history of their article (what does this mean?). If published, this will include your full peer review and any attached files.

Reviewer #1: No

Reviewer #3: **Yes: **Georg G. Duscher

---

## [Editor Report · Acceptance letter]

27 Jan 2021

Dear Prof. Otranto,

We are delighted to inform you that your manuscript, "Role of reptiles and associated arthropods in the epidemiology of rickettsioses: a one health paradigm," has been formally accepted for publication in PLOS Neglected Tropical Diseases.

Best regards,

Shaden Kamhawi

co-Editor-in-Chief

Paul Brindley

co-Editor-in-Chief
